# Traction-Associated Peridynamic Motion Equation and Its Verification in the Plane Stress and Fracture Problems

**DOI:** 10.3390/ma16062252

**Published:** 2023-03-10

**Authors:** Ming Yu, Zeyuan Zhou, Zaixing Huang

**Affiliations:** State Key Laboratory of Mechanics and Control of Mechanical Structures, Nanjing University of Aeronautics and Astronautics, 29 Yudao Street, Nanjing 210016, China

**Keywords:** peridynamics, traction-associated peridynamic motion equation, traction boundary condition, bond-based constitutive model

## Abstract

How to prescribe traction on boundary surface is still an open question in peridynamics. This problem is investigated in this paper. Through introducing the induced body force defined by boundary traction, the Silling’s peridynamic motion equation is extended to a new formulation called the traction-associated peridynamic motion equation, which is verified to be compatible with the conservation laws of linear momentum and angular momentum. The energy conservation equation derived from the traction-associated peridynamic motion equation has the same form as that in the original peridynamics advanced by Silling. Therefore, the constitutive models of the original peridynamics can be directly applied to the traction-associated peridynamic motion equation. Some benchmark examples in the plane stress problems are calculated. The numerical solutions agree well with the classical elasticity solutions, and the volume correction and the surface correction are no longer needed in the numerical algorithm. These results show that the traction-associated peridynamic motion equation not only retains all advantages of the original peridynamics, but also can conveniently deal with the complex traction boundary conditions.

## 1. Introduction

Peridynamics (PD) is a nonlocal continuum theory of mechanics developed in the recent two decades [1,2,3,4,5]. Its core consists in that a weighted integral of relative displacement over spatial domain is used instead of the gradient of displacement (strain) in the governing equations of deformation. Therefore, peridynamics can be used to conveniently and effectively analyze deformation companied with evolution of discontinuities. Peridynamics was firstly proposed by Silling [1] and then further improved by Silling and his collaborators [2]. Since then, it has been applied to investigate various problems associated with wave, damage, fracture, and impact breakage [3,4,5,6,7,8,9]. However, the traction boundary condition is incompatible with peridynamics because the governing equation of original peridynamics appears in the form of an integro-differential equation and does not involve the gradient with respect to spatial position. Therefore, boundary conditions cannot be imposed as naturally in the original peridynamics as in Classical Continuum Mechanics (CCM). A question therefore arises: how to incorporate suitable traction boundary condition?

Recalling the ways of the traction boundary condition is imposed in peridynamics, we can roughly divide them into three types. The first method [10] is to convert the traction on the boundary surface into the body force in an inner boundary layer according to the static force equivalence. The body force is usually supposed to uniformly distribute in the boundary layer, and the thickness of the inner boundary layer is taken as the spacing between material points. The second type of method is still based on the original peridynamics motion equation, but some modifications or operations will be performed to achieve the purpose of directly imposing traction boundary conditions. For example, a method only applicable to the non-ordinary correspondence models in peridynamics is proposed in [11], a weak form of peridynamic governing equations is proposed in [12,13], and a way based on Taylor expansion strategy to impose traction boundary conditions in ordinary state-based peridynamics is proposed in [14,15]. As the third method, Huang [16] suggested modifying the peridynamic motion equation to introduce the surface traction.

The traction boundary condition based on the first method is very popular in theoretical analysis and numerical calculation. Using this traction boundary condition, the well-posedness of the linear peridynamics with a given nonlocal kernel has been proved in mathematics [17,18]. Many numerical computation and analysis for practical problems can be found in [4,5,8]. However, it is unnatural and unhandy to convert the traction on the boundary surface to body forces in the inner boundary layer. Specially, when the material particles are close to the boundary surface or interface, the constitutive parameters need to be corrected [4,10,19,20]. Consequently, the traction boundary condition based on the first method is impractical for the sophisticated loading cases and geometrical surface.

If the non-ordinary correspondence models are adopted, the traction boundary condition can be specified directly. However, the non-ordinary correspondence models are limited due to the zero energy mode [21,22], and there are still some problems to be solved. The weak form of peridynamic governing equations has some changes comparing with the original peridynamic governing equation. Although the method of imposing traction boundary conditions based on Taylor expansion strategy can be successfully used in bond-based peridynamics, it still needs to set fictitious nodes. As for the model established by the third method, it is new and promising. Zhou [23] simplified the boundary transfer functions and proposed a method of imposing traction boundary conditions suitable for two-dimensional problems for bond-based peridynamics. However, since the three scalar-typical boundary transfer functions involved in this idea are difficult to determine, it has not been widely applied yet. Overall, how to characterize the traction boundary condition in peridynamics is still an open question and thus requires further investigation.

Meshfree discretization has been widely used to solve PD problems [24,25,26]. However, the standard discretization scheme is still very expensive for calculating the large-scale problems such as 3-dimensional problems. A new meshfree scheme [27] is expected to solve these difficulties.

In PD theory, there is no complete non-local neighborhood of a material point near a material boundary, resulting in the so-called skin or surface effects. Generally, the surface effect needs to be corrected to obtain the correct physical results. Various surface effect correction methods were studied in [10]. In addition, there are also many attempts to eliminate the surface correction [28,29,30,31,32,33].

The outline of the paper is as follows. In Section 2, through introducing the induced body force defined by boundary traction, we propose the traction-associated peridynamic motion equation and show that it is compatible with the conservation laws of linear momentum and angular momentum. In Section 3, the energy conservation law is derived from the traction-associated peridynamic motion equation. Two kinds of the bond-based constitutive models are discussed. The concrete form of the induced body force is determined. The prototype microelastic (PMB) constitutive model with the local damage is briefly introduced. The numerical algorithm is discussed in Section 4. In Section 5, the traction-associated peridynamic motion equation is used to calculate three benchmark examples in the plane stress problems. A fracture problem is simulated to verify the effectiveness of traction-associated peridynamic motion equation in the failure analysis. Finally, we close this paper with summary and comment.

## 2. The Induced Body Force and Extension of Peridynamic Motion Equation

Let **p**(**x**′′′, *t*) is a traction exerted at the point **x**′′′ on the boundary surface *∂*Ω*_p_* ⊂ *∂*Ω of a peridynamic media Ω. Due to the nonlocality of peridynamic media, **p**(**x**′′′, *t*) will permeate the interior of Ω and induce a body force **b_i_**(**x**, *t*) acting at the material particles. We call **b_i_**(**x**, *t*) the induced body force, which is represented as
(1)$$b_{i}\left(x,t\right)=\int_{\partial \Omega_{p}}\left[G\left(x,x^{\prime \prime \prime },t\right)\cdot p\left(x^{\prime \prime \prime },t\right)\right]dA_{x^{\prime \prime \prime }}.$$

In Equation (1), **G** (**x**, **x**′′′, *t*) is a second-order tensor field called the transfer function of boundary traction and with the dimension of 1/m^3^, which reads
(2)$$G\left(x,x^{\prime \prime \prime },t\right)=P\left(x,x^{\prime \prime \prime },t\right)\cdot S\left(x^{\prime \prime \prime },t\right),$$
where
(3)$$\left\{\begin{array}{l} P\left(x,x^{\prime \prime \prime },t\right)=\alpha \left(x^{\prime \prime \prime },x\right)\frac{\left(y^{\prime \prime \prime }-y\right)⊗\left(y^{\prime \prime \prime }-y\right)}{\left(y^{\prime \prime \prime }-y\right)\cdot \left(y^{\prime \prime \prime }-y\right)}\\ S\left(x^{\prime \prime \prime },t\right)={\left[\int_{\Omega }P\left(x,x^{\prime \prime \prime },t\right)dV_{x}\right]}^{-1} \end{array}\right..$$

In Equation (3), **y**′′′ = **x**′′′ + **u**′′′(**x**′′′, *t*) is the position vector of **x**′′′ in the deformed configuration and **u**′′′ = **u**′′′(**x**′′′, *t*) is a displacement field on *∂*Ω*_p_*. *α*(**x**′′′, **x**) is a weight function determining the characteristic of **G**(**x**, **x**′′′, *t*) and will be discussed in the following. By Equations (2) and (3), it is easy to verify
(4)$$\int_{\Omega }G\left(x,x^{\prime \prime \prime },t\right)dV_{x}=I,$$
where **I** is the second order unit tensor. Equation (4) is a sufficient and necessary condition to ensure the compatibility of peridynamic motion equation with total equilibrium of linear momentum and angular momentum. Next, let us discuss this argument.

By Equation (4), the integrals of Equation (1) and **y**(**x**, *t*) × Equation (1) over Ω lead to
(5)$$\int_{\Omega }b_{i}\left(x,t\right)dV_{x}=\int_{\partial \Omega_{p}}p\left(x^{\prime \prime \prime },t\right)dA_{x^{\prime \prime \prime }}$$
(6)$$\int_{\Omega }y\left(x\right)\times b_{i}\left(x,t\right)dV_{x}=\int_{\partial \Omega_{p}}y^{\prime \prime \prime }\left(x^{\prime \prime \prime }\right)\times p\left(x^{\prime \prime \prime },t\right)dA_{x^{\prime \prime \prime }}.$$

After the induced body force **b_i_**(**x**, *t*) is introduced, the motion equation of peridynamic media subjected simultaneously to boundary traction and external body force can be written as
(7)$$\rho \left(x\right)\ddot{u}\left(x,t\right)=b_{i}\left(x,t\right)+\int_{H_{x}}\left\{Ṯ\left[x,t\right]\langle \xi \rangle -Ṯ\left[x^{\prime },t\right]\langle -\xi \rangle \right\}dV_{x^{\prime }}+b\left(x,t\right),$$
where **ξ** = **x**′ − **x**, *H***_x_** is a spherical neighborhood of **x** with radius *δ*, **T** the force vector state field [2], **b**(**x**, *t*) the external body force, and *ρ*(**x**) the mass density. We call Equation (7) the traction-associated peridynamic motion equation. Clearly, Equation (7) is an extension of the Silling’s peridynamic motion equation. Let **x**′′′∈*∂*Ω*_p_* and **x**∈Ω. If we take **G**(**x**, **x**′′′, *t*) = H(|**x**′′′−**x**|)**I**/V*_b_* where H(|**x**′′′ − **x**|) is the dimensionless square wave function and V*_b_* the volume of the boundary layer of *∂*Ω*_p_* with the thickness of Δ, then Equation (1) will degenerate into a common formula when ones deal with the boundary traction in peridynamics.

By Equations (5) and (6), the integrals of Equation (7) and **y**(**x**, *t*) × Equation (7) over Ω yield
(8)$$\int_{\Omega }\rho \left(x\right)\ddot{u}\left(x,t\right)dV_{x}=\int_{\partial \Omega_{p}}p\left(x,t\right)dA_{x}+\int_{\Omega }b\left(x,t\right)dV_{x}$$
(9)$$\int_{\Omega }\rho \left(x\right)\overset{\cdot }{\bar{y\left(x\right)\times v\left(x,t\right)}}dV_{x}=\int_{\partial \Omega_{p}}y\left(x\right)\times p\left(x,t\right)dA_{x}+\int_{\Omega }y\left(x\right)\times b\left(x,t\right)dV_{x},$$
which describe total equilibrium of linear momentum and angular momentum, respectively. Therefore, Equation (7) is consistent with the conservation laws of linear momentum and angular momentum. In addition, it is easy to see that Equation (7) is form-invariant under the Galileo transformation.

## 3. Peridynamic Constitutive Model

### 3.1. Balance Equation of Energy

Let **v** = **v**(**x**) is the velocity field within material, and *ε* is the internal energy density. Only elastic deformation is concerned, in peridynamics, total energy conservation can be represented as
(10)$$\frac{D}{Dt}\int_{\Omega }\left(\frac{1}{2}\rho v^{2}+\rho \varepsilon \right)dV=\int_{\Omega }b_{i}\cdot vdV_{x}+\int_{\Omega }b\cdot vdV.$$

Equation (10) can be further written as
(11)$$\int_{\Omega }\rho a\cdot vdV+\int_{\Omega }\rho \dot{\varepsilon }dV=\int_{\Omega }b_{i}\cdot vdV_{x}+\int_{\Omega }b\cdot vdV,$$
where **a** is acceleration field. In terms of Equation (7), Equation (11) reduces to
(12)$$\int_{\Omega }\rho \dot{\varepsilon }dV=\int_{\Omega }\left\{\int_{H_{x}}\left\{Ṯ\left[x^{\prime },t\right]\langle -\xi \rangle -Ṯ\left[x,t\right]\langle \xi \rangle \right\}dV_{x^{\prime }}\right\}\cdot vdV_{x}.$$

Since *H***_x_** ⊂ Ω is a compact supported set of **T**[**x**′, *t*]<−**ξ**> and **T**[**x**, *t*]<**ξ**>, Equation (12) can be written as
(13)$$\int_{\Omega }\rho \dot{\varepsilon }dV=\int_{\Omega }\left\{\int_{\Omega }\left\{Ṯ\left[x^{\prime },t\right]\langle -\xi \rangle -Ṯ\left[x,t\right]\langle \xi \rangle \right\}dV_{x^{\prime }}\right\}\cdot vdV_{x}.$$

Exchanging **x**′ and **x**, and then using definition of the compact supported set, we have
(14)$$\int_{\Omega }\rho \dot{\varepsilon }dV=\int_{\Omega }\left\{\int_{H_{x}}Ṯ\left[x,t\right]\langle \xi \rangle \left[v\left(x^{\prime }\right)-v\left(x\right)\right]dV_{x^{\prime }}\right\}dV_{x}.$$

By the localized hypothesis [34,35], the balance equation of energy is given as follows
(15)$$\rho \dot{\varepsilon }=\int_{H_{x}}Ṯ\left[x,t\right]\langle \xi \rangle \left[v\left(x^{\prime }\right)-v\left(x\right)\right]dV_{x^{\prime }}.$$

Clearly, Equation (15) has the same form as that in the original peridynamics advanced by Silling and his collaborators [1,2]. Equation (15) is a basis to determine the peridynamic constitutive models of hyperelastic material. Therefore, the hyperelastic constitutive models in the original peridynamics can be inherited without modification by the traction-associated peridynamics.

### 3.2. Bond-Based Constitutive Models

Bond-based (BB) constitutive models are simplified versions of state-based (SB) constitutive models. For brevity, only the BB constitutive models are considered, not concerned the SB constitutive models. The BB constitutive models have been systematically established by Silling [1], and their forms are not unique. The microelastic models [5] are used to describe the elastic deformation of isotropic materials. Two commonly used microelastic models are listed below.

General microelastic models:


(16)
$$f\left(u^{\prime }-u,\xi \right)=\left\{\begin{array}{ll} c\left(\left|y^{\prime }-y\right|-\left|\xi \right|\right)\frac{y^{\prime }-y}{\left|y^{\prime }-y\right|}\; & \left|\xi \right|\leq \delta \\ 0 & \mathrm{otherwise} \end{array}\right..$$


The linear form of Equation (16) is
(17)$$f\left(u^{\prime }-u,\xi \right)=\left\{\begin{array}{ll} c\frac{\xi ⊗\xi }{{\left|\xi \right|}^{2}}\cdot \left(u^{\prime }-u\right)\; & \left|\xi \right|\leq \delta \\ 0 & \mathrm{otherwise} \end{array}\right.,$$
where **f**(**u**′−**u**, **ξ**) is the force density vector [1] with the dimension of N/m^6^. The relation between **f**(**u**′−**u**, **ξ**) and the force vector state **T** in Equation (7) is represented as **T**[**x**, *t*] <**ξ**> = −**T** [**x**′, *t*] <−**ξ**> == **f**(**u**′−**u**, **ξ**)/2. The parameter *c* is called the spring constant or the bond-constant, which reads [5]
(18)$$c=\left\{\begin{array}{ll} \frac{15E}{\pi \delta^{5}}\; & 3-\mathrm{dimension}\\ \frac{12E}{\pi h\delta^{4}}\; & 2-\mathrm{dimension}\;\mathrm{plane}\;\mathrm{stress}\\ \frac{64E}{5\pi h\delta^{4}}\; & 2-\mathrm{dimension}\;\mathrm{plane}\;\mathrm{strain}\\ \frac{3E}{h_{1}\delta^{3}}\; & 1-\mathrm{dimension} \end{array}\right.,$$
where *E* is the Young’s modulus, *h* the thickness of plate and *h*_1_ the cross-sectional area of rod.

2.The prototype microelastic (PM) model:

PM model is another special form of the microelastic models
(19)$$f\left(u^{\prime }-u,\xi \right)=\left\{\begin{array}{ll} c\frac{\left|y^{\prime }-y\right|-\left|\xi \right|}{\left|\xi \right|}\frac{y^{\prime }-y}{\left|y^{\prime }-y\right|}\; & \left|\xi \right|\leq \delta \\ 0 & \mathrm{otherwise} \end{array}\right..$$

The linear form of Equation (19) is
(20)$$f\left(u^{\prime }-u,\xi \right)=\left\{\begin{array}{ll} c\frac{\xi ⊗\xi }{{\left|\xi \right|}^{3}}\cdot \left(u^{\prime }-u\right)\; & \left|\xi \right|\leq \delta \\ 0 & \mathrm{otherwise} \end{array}\right..$$

The parameter *c* in Equations (19) and (20) is still the bond-constant and it takes the value below [4,5]
(21)$$c=\left\{\begin{array}{ll} \frac{12E}{\pi \delta^{4}}\; & 3-\mathrm{dimension}\\ \frac{9E}{\pi h\delta^{3}}\; & 2-\mathrm{dimension}\;\mathrm{plane}\;\mathrm{stress}\\ \frac{48E}{5\pi h\delta^{3}}\; & 2-\mathrm{dimension}\;\mathrm{plane}\;\mathrm{strain}\\ \frac{2E}{h_{1}\delta^{2}}\; & 1-\mathrm{dimension} \end{array}\right..$$

It should be noted that in the BB models, the Poisson’s ratio of 3D and 2D plane strain problems are fixed at 1/4, while that of 2D plane stress problem are fixed at 1/3.

### 3.3. Transfer Function of Boundary Traction

According to Equations (2) and (3), the transfer function **G**(**x**, **x**′′′, *t*) of the boundary traction characterizes the change of intensity when the traction is transferred from the boundary surface to the interior of the body, which is determined by the weight function *α*(**x**′′′, **x**). In physics, *α*(**x**′′′, **x**) should attenuates to zero as it moves away from the boundary surface. Therefore, *α*(**x**′′′, **x**) can be defined a function with compact support, i.e.,
(22)$$\alpha \left(x^{\prime \prime \prime },x\right)=\left\{\begin{array}{ll} q\left(\left|x^{\prime \prime \prime }-x\right|\right)\neq 0\; & \left|x^{\prime \prime \prime }-x\right|\leq \iota \\ 0 & \mathrm{otherwise} \end{array}\right.,$$
where **x**′′′ ∈ *∂*Ω*_p_* while **x** ∈ Ω. *ι* is a scale parameter. For simplicity, we take *ι* = *δ*. As thus, Equation (22) means that the traction on the boundary surface is dispersed in the boundary layer with the thickness of *δ*.

Consider the quasi-static uniaxial tension of a rod subjected to tensile force *p* at two ends. Through the inverse method [36] and the undetermined coefficient method, we find that when the general microelastic constitutive Equations (16) and (17) are adopted, if *α*(*x*′′′, *x*) takes the form below
(23)$$\alpha \left(x^{\prime \prime \prime },x\right)=\left\{\begin{array}{ll} k\left(\delta^{2}-{\left|x^{\prime \prime \prime }-x\right|}^{2}\right)\; & \left|x^{\prime \prime \prime }-x\right|\leq \delta \\ 0 & \mathrm{otherwise} \end{array}\right.$$
then the classical elasticity solution of the uniaxial tension can be acquired.

Similarly, when the PM constitutive Equations (19) and (20) are used, if *α*(*x*′′′, *x*) is written as
(24)$$\alpha \left(x^{\prime \prime \prime },x\right)=\left\{\begin{array}{ll} k\left(\delta -\left|x^{\prime \prime \prime }-x\right|\right)\; & \left|x^{\prime \prime \prime }-x\right|\leq \delta \\ 0 & \mathrm{otherwise} \end{array}\right.$$
the same result is also given. It should be noted that *k* in Equations (23) and (24) is any constant and is not zero. For simplicity, we can take *k* = 1.

Equations (23) and (24) can be extended to 2-dimensional and 3-dimensional form below
(25)$$\alpha \left(x^{\prime \prime \prime },x\right)=\left\{\begin{array}{ll} \delta^{2}{\left|x^{\prime \prime \prime }-x\right|}^{2}\; & \left|x^{\prime \prime \prime }-x\right|\leq \delta \\ 0 & \mathrm{otherwise} \end{array}\right.$$
(26)$$\alpha \left(x^{\prime \prime \prime },x\right)=\left\{\begin{array}{ll} \delta -\left|x^{\prime \prime \prime }-x\right|\; & \left|x^{\prime \prime \prime }-x\right|\leq \delta \\ 0 & \mathrm{otherwise} \end{array}\right..$$

Thus, the concrete form of the induced body force is determined due to Equation (25) or (26), In the following, they will be directly used to analyze benchmark examples of 2D plane stress problems.

### 3.4. Prototype Microelastic Brittle Damage Model

In PD theory, local damage at a point is defined as the weighted ratio of the number of eliminated interactions to the total number of initial interactions of the material point with its family members, that is [4,19]
(27)$$\varphi \left(x,t\right)=1-\frac{\int_{H_{x}}\mu \left(\xi ,t\right)dV_{x^{\prime }}}{\int_{H_{x}}dV_{x^{\prime }}}.$$

It should be noted that the local damage *φ* ranges from 0 to 1. When *φ* = 1, all the interactions initially associated with the point have been eliminated, while *φ* = 0 means that all interactions exist. The measurement of the local damage value is an indicator of the possible formation of cracks within a body. In Equation (27), *μ* is a history-dependent scalar-valued function, which reads
(28)$$\mu \left(\xi ,t\right)=\left\{\begin{array}{ll} 1\; & s\left(\xi ,t^{\prime }\right)<S_{c}\;for\;all\;0\leq t^{\prime }\leq t\\ 0 & \mathrm{otherwise} \end{array}\right.,$$
where *S_c_* is the critical stretch of bond failure, while *s* is bond stretch defined by
(29)$$s=\frac{\left|y^{\prime }-y\right|-\left|\xi \right|}{\left|\xi \right|}.$$

Therefore, a simple way to introduce failure into the constitutive model to allow bonds (springs) to break when they are stretched beyond a predefined limit. After bond failure, there is no tensile force sustainable in the bond, and once a bond fails, it is failed forever (there is no provision for “healing” of a failed bond). As thus, the PMB model to characterize brittle damage [4,5,19] can be written as
(30)$$f\left(u^{\prime }-u,\xi ,t\right)=\left\{\begin{array}{ll} cs\mu \left(\xi ,t\right)\frac{y^{\prime }-y}{\left|y^{\prime }-y\right|}\; & \left|\xi \right|\leq \delta \\ 0 & \mathrm{otherwise} \end{array}\right..$$

## 4. Numerical Algorithm

### 4.1. Spatial Discretization

Meshfree spatial discrete method [19] is used to discretize continuum into a range of arbitrary shaped subdomains, in which, the collocation points (nodes) are placed. With one-point Gauss quadrature strategy [37], the spatial discrete form of Equation (7) can be written as
(31)$$\begin{array}{l} \rho \left(x_{i}\right)\ddot{u}\left(x_{i},t\right)\\ =\sum_{\partial \Omega_{p}}G\left(x_{i},x_{k},t\right)\cdot \bar{p}\left(x_{k},t\right)A_{x_{k}}+\sum_{H_{x_{i}}}\left\{Ṯ\left[x_{i},t\right]\langle x_{j}-x_{i}\rangle -Ṯ\left[x_{j},t\right]\langle x_{i}-x_{j}\rangle \right\}V_{x_{j}}+b\left(x_{i},t\right). \end{array}$$

The same spatial discrete strategy can be applied to acquire the integral value of **G**(**x**, **x**′′′). It is worth noting that the collocation points must also be set on the traction boundary surface due to the introduction of the boundary integration.

### 4.2. Time Integration

The adaptive dynamic relaxation (ADR) method [38] is used to solve Equation (31). After introducing new fictitious inertia and damping terms, Equation (31) is rewritten as
(32)$$\lambda_{i}{\ddot{u}}_{i}^{n}+c^{n}\lambda_{i}{\dot{u}}_{i}^{n}=F_{i}^{n},$$
where *n* is *n*-th time (iterative) step, *λ_i_* the fictitious density of the *i*-th node, and *c^n^* the damping coefficient. The vector **F***_i_^n^* in Equation (32) is the summation of internal and external forces, which reads
(33)$$\begin{array}{l} F_{i}^{n}\\ =\sum_{\partial \Omega_{p}}G\left(x_{i},x_{k},t\right)\cdot \bar{p}\left(x_{k},t\right)A_{x_{k}}+\sum_{H_{x_{i}}}\left\{Ṯ\left[x_{i},t\right]\langle x_{j}-x_{i}\rangle -Ṯ\left[x_{j},t\right]\langle x_{i}-x_{j}\rangle \right\}V_{x_{j}}+b\left(x_{i},t\right). \end{array}$$

By the explicit central-difference integration, the iterative scheme of displacements and velocities can be written as
(34)$$\left\{\begin{array}{l} {\dot{u}}_{i}^{n+\frac{1}{2}}=\frac{\left(2-c^{n}\Delta t\right){\dot{u}}_{i}^{n-\frac{1}{2}}+\frac{2\Delta tF_{i}^{n}}{\lambda_{i}}}{2+c^{n}\Delta t}\\ u_{i}^{n+1}=u_{i}^{n}+\Delta t{\dot{u}}_{i}^{n+\frac{1}{2}} \end{array}\right..$$

As Equation (34) has an unknown velocity field at *t*^−1/2^, the iterative process can be not started. However, if we assume **u***_i_*^0^ ≠ **0** and **v***_i_*^0^ = **0**, the iteration can be completed by using
(35)$${\dot{u}}_{i}^{\frac{1}{2}}=\frac{\Delta tF_{i}^{0}}{2\lambda_{i}}.$$

In this algorithm, since the fictitious density *λ_i_*, the damping coefficient *c^n^* and the time step Δ*t* are not actual physical quantities, their values can be chosen so as to make the convergence of numerical solution as fast as possible. Therefore, we take Δ*t* = 1, while the calculation of *λ_i_* and *c^n^* can refer to [4,38,39,40].

## 5. Some Plane Stress Benchmark Problems

The nonlinear PM constitutive model is adopted to analyze some benchmark examples of the plane stress problems. In calculation, we take the Young’s modulus *E* = 200 GPa and the Poisson’s ratio *ν* = 1/3.

### 5.1. Example 1: A Rectangular Plate with Two Opposite Edges Subjected to Tension

As shown in Figure 1, the upper and lower edges of an isotropic rectangular plate are subjected to uniform tension *q* = 200 MPa. The length of the plate *L* = 1 m, the width *W* = 0.5 m, and the thickness *h* = Δ.

The rectangular plate (excepting near the boundary) is uniformly discretized into a particle set with the equal spacing Δ in the plane, as shown in Figure 2.

In Figure 2, the black dots represent the collocation points (nodes). Due to the symmetry of the structure and load, the constrains to rigid-body displacement need be imposed at the two symmetrical axes of the plate. It is necessary to collocate nodes on the boundary. The volume of subdomain associated with the boundary node is Δ^2^*h*/2 or Δ^2^*h*/4.

When the horizon size is specified as *δ* = 3.015Δ, three different grid sizes Δ = *L*/50, *L*/100, and *L*/200 are used to show the influence of the grid density on the convergence and computational accuracy, as illustrated in Figure 3 and Figure 4.

It can be seen from Figure 3 that the numerical calculation converges at the 500th time step regardless of the grid size. Moreover, Figure 4 shows that the computational results are close to each other for different grid density, and the errors between them and classical solutions are all within 5%. Therefore, the numerical algorithm can be considered to satisfy the *δ*-convergence requirement [41,42,43,44].

When the horizon size is fixed as *δ* = *m*Δ = 0.06 m, three different combinations (Δ = *L*/50, *m* = 3; Δ = *L*/100, *m* = 6; Δ = *L*/200, *m* = 12) are used to study *m*-convergence. The results are illustrated in Figure 5 and Figure 6.

Figure 5 that the calculation converges very quickly with time step, no matter which combination mode is adopted. From Figure 6, we see that the numerical results of the three different combinations agree with each other, and the errors between them and classical solutions are all within 5%. Therefore, the numerical algorithm is of *m*-convergence [41,42,43,44].

Comparison between Equation (7) and the original PD is carried out through numerical calculation. Figure 3 and Figure 7 show that the displacements calculated by Equation (7) and the original PD converges very quickly with time step, and the convergence modes are similar. As can be seen from Figure 8, Figure 9 and Figure 10, there exist good matches between predictions of traction-associated peridynamic motion equation and analytical solutions as well as original PD predictions.

In order to balance the computational accuracy and efficiency, in the following, the horizon size is specified as *δ* = 3.015Δ and grid size Δ = *L*/100. The total time step is set to 3000. The material parameters of plates in all examples are the same as those of the plate in Figure 1, and the same discretization as Figure 2 is adopted.

### 5.2. Example 2: A Rectangular Plate Subjected to Bending

As shown in Figure 11, a rectangular plate with the same size as the plate in Figure 1 is subjected to an anti-symmetrically distributed loads with a maximal value *q* = 200 Mpa. This is a pure bending problem with the stress boundary condition.

The convergence analysis is shown in Figure 12. It can be seen that the displacements calculated by the original PD and Equation (7) converge with time step in the same way, and completely converge at the 1000th time step.

The numerical results are illustrated in Figure 13, Figure 14, Figure 15, Figure 16 and Figure 17, From which, it can be seen that the displacement distribution predicted by Equation (7) agrees with that by original PD, and the two have better matching with analytical solutions.

### 5.3. Example 3: A Square Plate with A Circular Hole Subjected to Tension by Two Opposite Edges

As shown in Figure 18, a squared plate with central circular hole is subjected to uniform tension *q* = 200 MPa. The side length of the plate *L* = 0.5 m and the radius of the circular hole *r* = 0.05 m.

The convergence analysis is shown in Figure 19. It can be seen that the displacements calculated by the original PD and Equation (7) converge with time step in the same way, and completely converge at the 500th time step.

The results in Figure 20 show that the displacements given by FEA, PD, and Equation (7) are close to each other in distribution. The relative error between them is less than 6.7%.

### 5.4. Example 4: Failure of A Square Plate with A Circular Hole under Quasi-Static Loading

We continue to investigate the fracture of the plate with central circular hole under tension. As shown in Figure 21, in order to avoid nodes occurring at the propagation path of cracks, all nodes (red and black dots) have been anew collocated. The grid size is taken as Δ = *L*/100. The nodes on the boundary surface (red dots) are only involved in the integration of the boundary traction and correspond to a volume of 0. In calculation, the critical stretch *S_c_* of bond failure takes 0.0058.

The propagation of crack is characterized by the value of the damage *φ*. When the load *q* arrives at 380 MPa, the plate breaks due to cracking. Figure 22 shows the damage *φ* calculated by the original PD. At the 500th time step, the crack initiates from two sides of the hole. As the time step increases to 700, the crack propagates and the damage *φ* reaches 0.479. As the time step continues to increase, the plate breaks at the 900th time step.

The damage *φ* calculated by Equation (7) is illustrated in Figure 23. The results show that the crack initiates at the 700th time step, and at the 900th time step, the damage *φ* of the crack propagation reaches 0.42. When the time step arrives at 1125, the plate fails due to cracking.

## 6. Conclusions

Through introducing the induced body force defined by boundary traction, the Silling’s peridynamic motion equation is extended to the traction-associated peridynamic motion equation. From investigation on this equation, the conclusions are summarized as follows.

The traction-associated peridynamic motion equation is consistent with the conservation laws of linear and angular momentum, and it is form-invariant under the Galileo transformation.The constitutive models in the original peridynamics can be inherited without modification by the traction-associated peridynamics. The concrete form of the induced body force is determined by matching with the constitutive models.Numerical calculations for the typical plane stress problems are in good agreement with the classical elasticity solutions, and the volume correction and the surface correction are no longer needed in the numerical algorithm.

## Figures and Tables

**Figure 1 materials-16-02252-f001:**
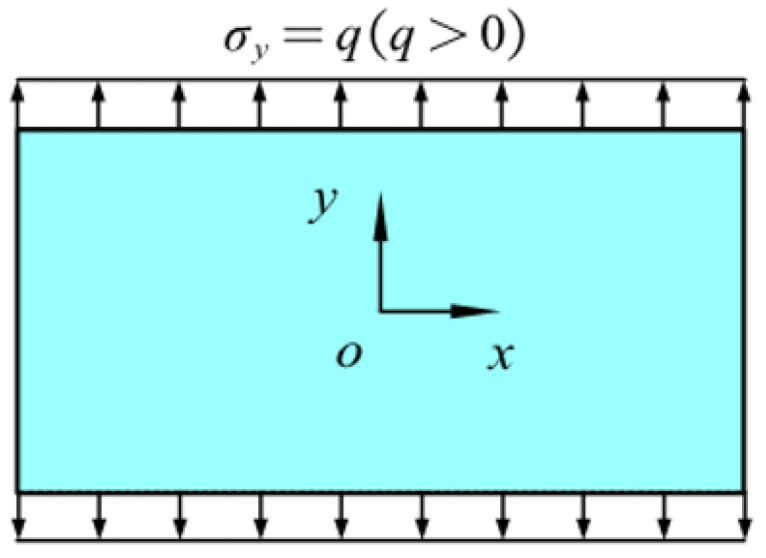
The rectangular plate subjected to uniform tension.

**Figure 2 materials-16-02252-f002:**
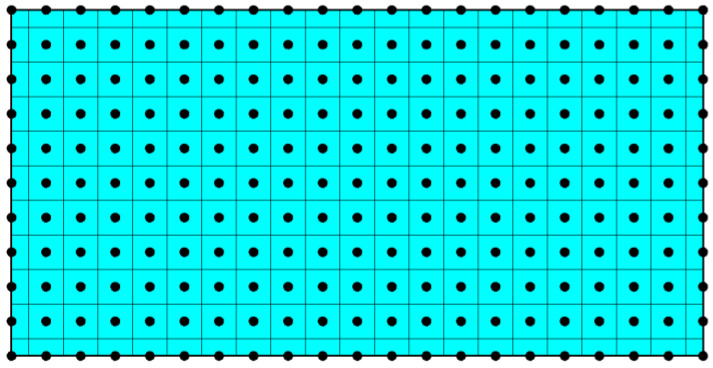
The spatial discretization of the rectangular plate.

**Figure 3 materials-16-02252-f003:**
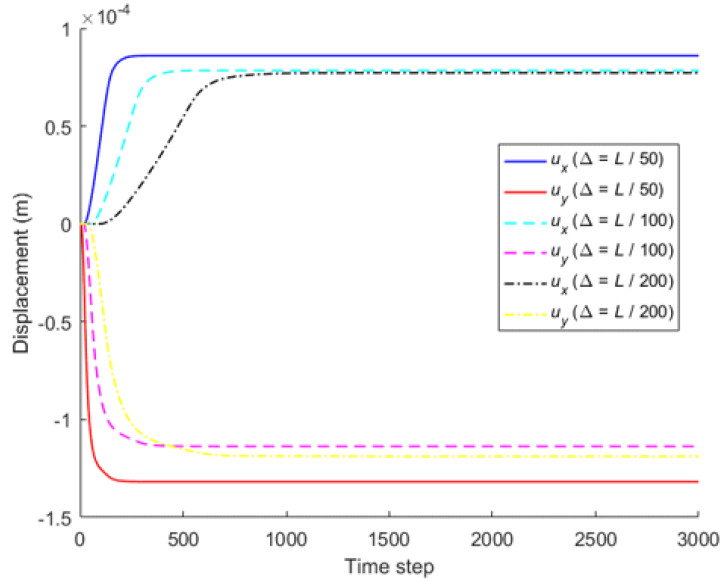
*δ*-convergence of displacement with time step at different nodes.

**Figure 4 materials-16-02252-f004:**
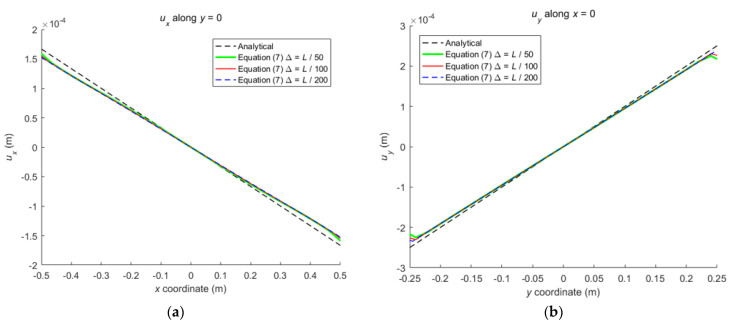
The displacement along central lines of the rectangular plate: (**a**) *u_x_* along *y* = 0; (**b**) *u_y_* along *x* = 0.

**Figure 5 materials-16-02252-f005:**
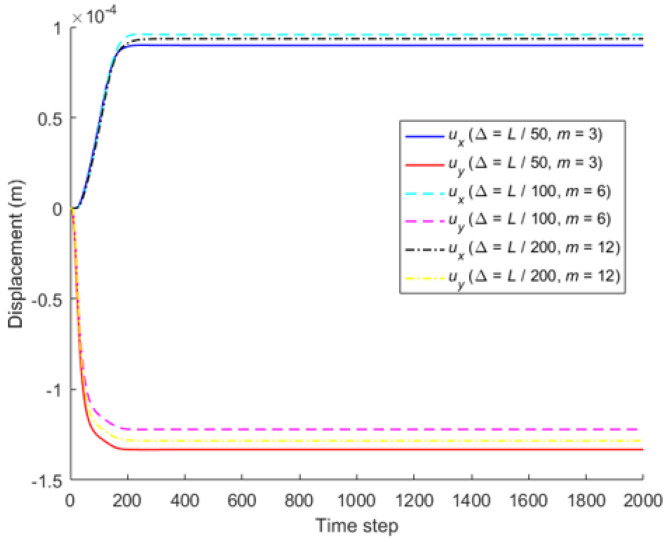
*m*-convergence of displacement with time step at different nodes.

**Figure 6 materials-16-02252-f006:**
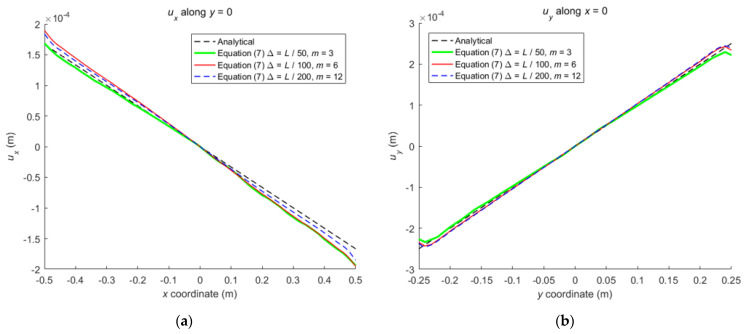
The displacements along central lines of the rectangular plate for three different combinations: (**a**) *u_x_* along *y* = 0; (**b**) *u_y_* along *x* = 0.

**Figure 7 materials-16-02252-f007:**
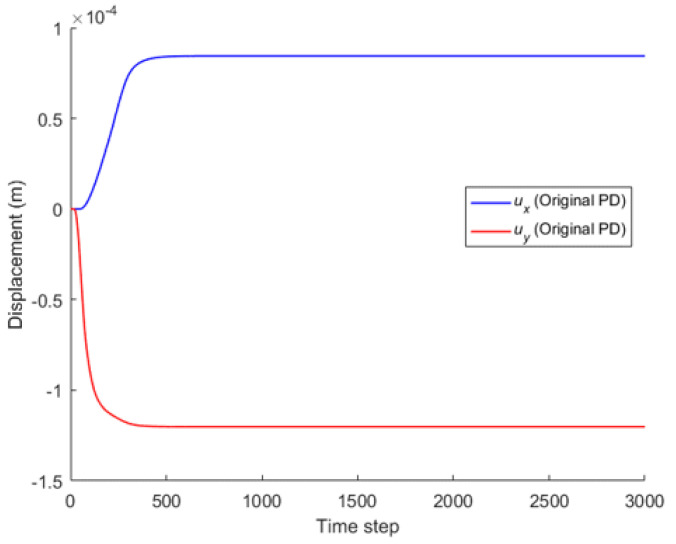
Convergence of original PD with time step.

**Figure 8 materials-16-02252-f008:**
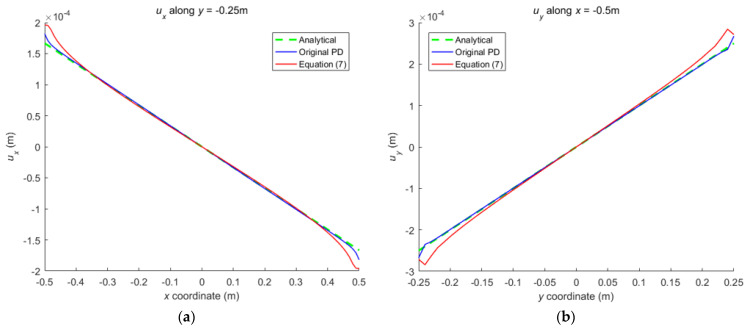
The displacement along the bottom side and left side of the rectangular plate subjected to tension: (**a**) *u_x_* along *y* = −0.25 m; (**b**) *u_y_* along *x* = −0.5 m.

**Figure 9 materials-16-02252-f009:**
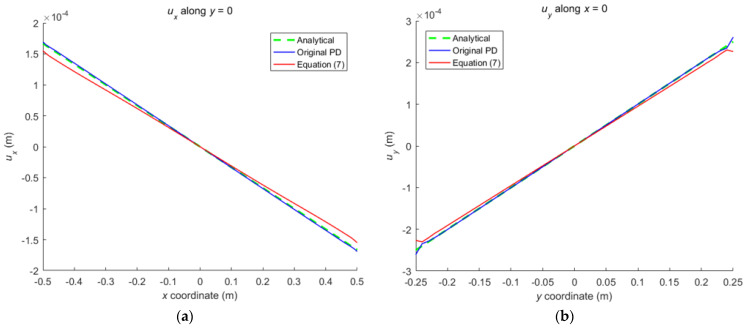
The displacement along central lines of the rectangular plate subjected to tension: (**a**) *u_x_* along *y* = 0; (**b**) *u_y_* along *x* = 0.

**Figure 10 materials-16-02252-f010:**
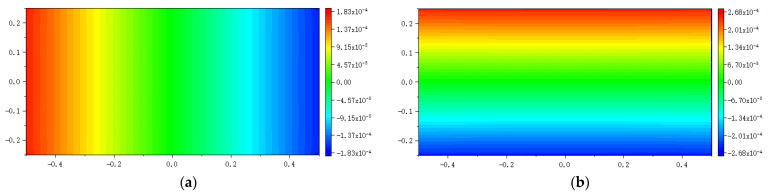
The distribution of displacement in the rectangular plate subjected to tension: (**a**) *u_x_* calculated by original PD; (**b**) *u_y_* calculated by original PD; (**c**) *u_x_* calculated by Equation (7); (**d**) *u_y_* calculated by Equation (7).

**Figure 11 materials-16-02252-f011:**
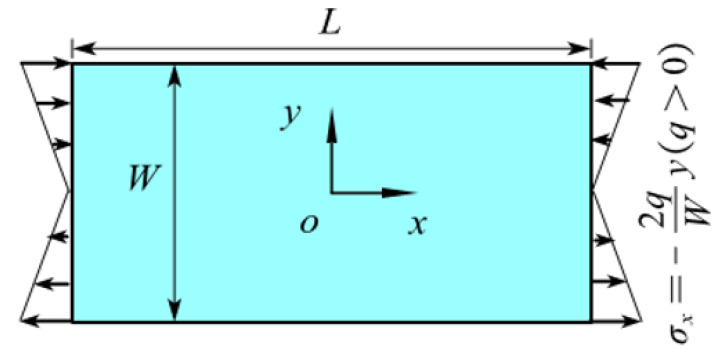
The plate subjected to bending.

**Figure 12 materials-16-02252-f012:**
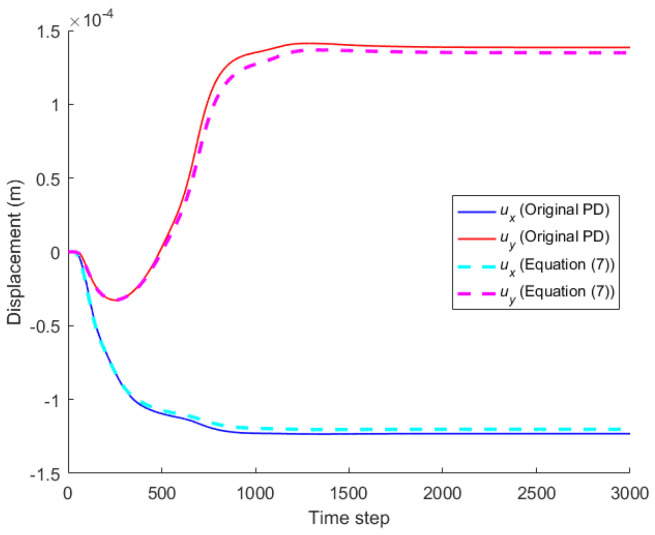
Convergence of displacement at *x* = −0.25 m and *y* = −0.12 m with time step.

**Figure 13 materials-16-02252-f013:**
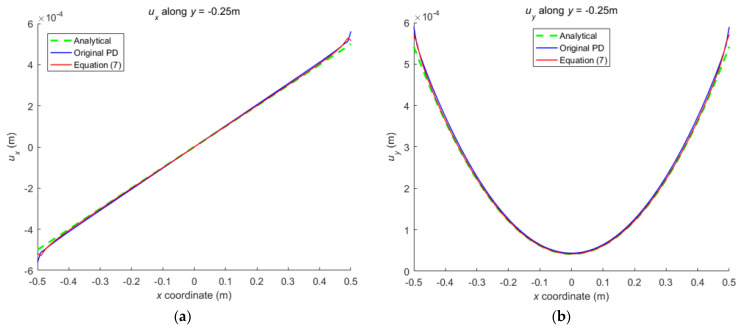
The displacement along the bottom side when the plate bending: (**a**) *u_x_* along *y* = −0.25 m; (**b**) *u_y_* along *y* = −0.25 m.

**Figure 14 materials-16-02252-f014:**
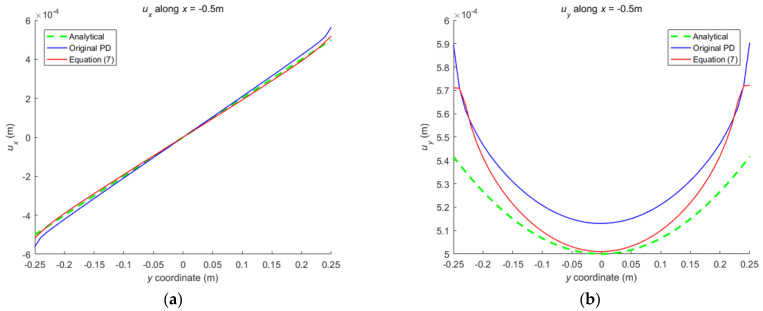
The displacement along the left side when the plate bending: (**a**) *u_x_* along *x* = −0.5 m; (**b**) *u_y_* along *x* = −0.5 m.

**Figure 15 materials-16-02252-f015:**
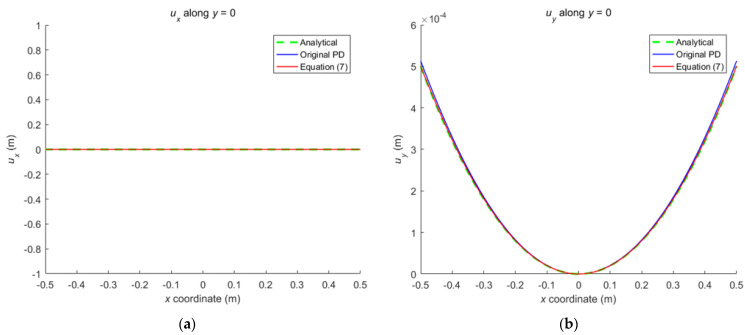
The displacement along the horizontal line *y* = 0 when the plate bending: (**a**) *u_x_* along *y* = 0; (**b**) *u_y_* along *y* = 0.

**Figure 16 materials-16-02252-f016:**
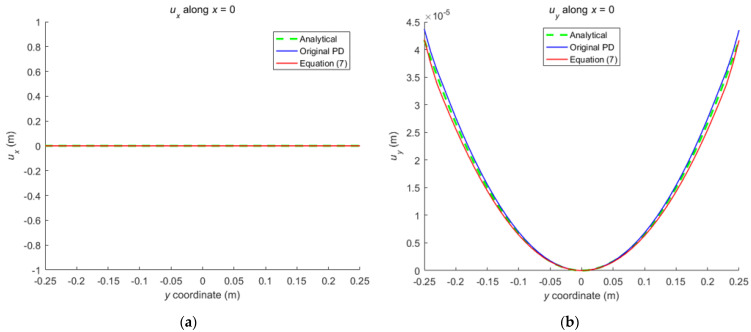
The displacement along the vertical line *x* = 0 when the plate bending: (**a**) *u_x_* along *x* = 0; (**b**) *u_y_* along *x* = 0.

**Figure 17 materials-16-02252-f017:**
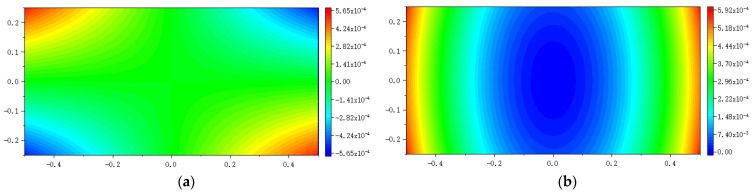
The distribution of displacement when the plate bending: (**a**) *u_x_* calculated by original PD; (**b**) *u_y_* calculated by original PD; (**c**) *u_x_* calculated by Equation (7); (**d**) *u_y_* calculated by Equation (7).

**Figure 18 materials-16-02252-f018:**
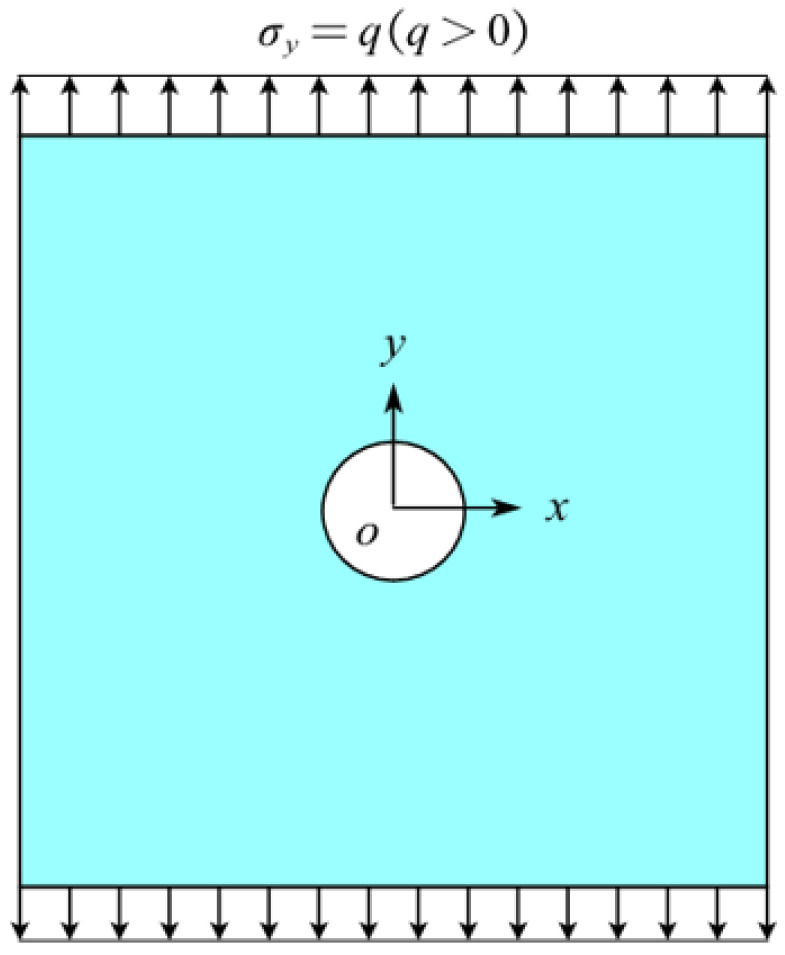
Squared plate with central circular hole subjected to uniform tension.

**Figure 19 materials-16-02252-f019:**
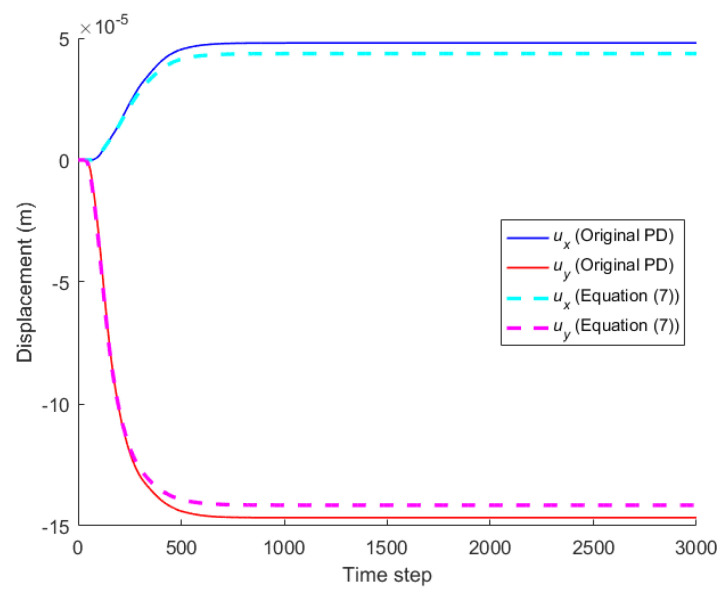
Convergence of displacement at *x* = −0.125 m and *y* = −0.125 m with time step.

**Figure 20 materials-16-02252-f020:**
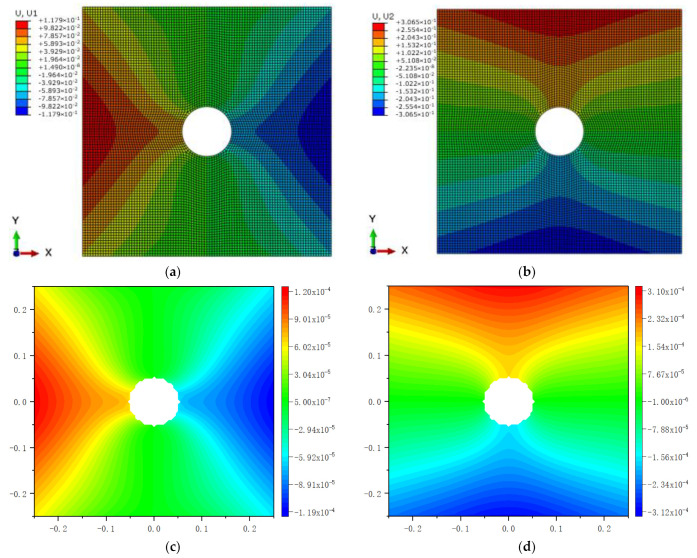
The distribution of displacement in the plate with central circular hole: (**a**) *u_x_* calculated by ABAQUS; (**b**) *u_y_* calculated by ABAQUS; (**c**) *u_x_* calculated by original PD; (**d**) *u_y_* calculated by original PD; (**e**) *u_x_* calculated by Equation (7); (**f**) *u_y_* calculated by Equation (7).

**Figure 21 materials-16-02252-f021:**
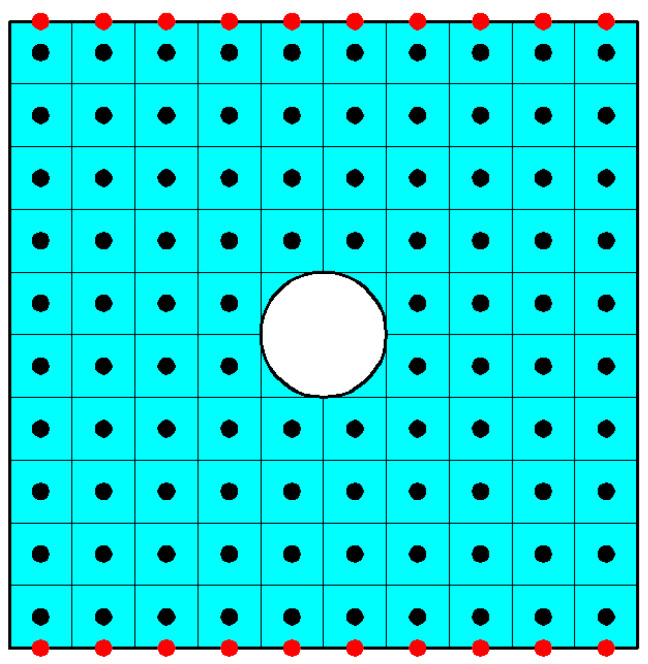
The discretization of the square plate.

**Figure 22 materials-16-02252-f022:**
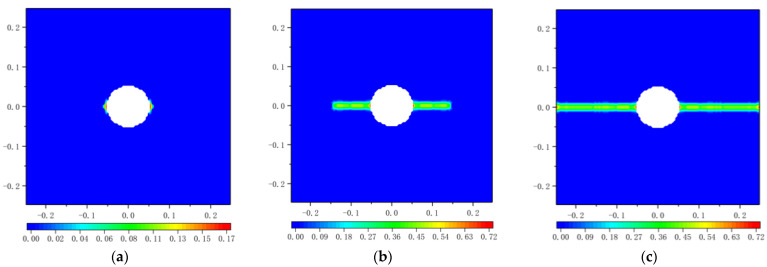
Damage plots for the square plate with a circular hole at the end of different time steps based on original PD: (**a**) 500 time steps; (**b**) 700 time steps; (**c**) 900 time steps.

**Figure 23 materials-16-02252-f023:**
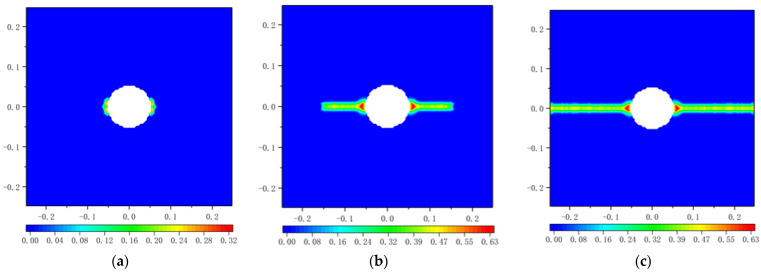
Damage plots for the square plate with a circular hole at the end of different time steps based on Equation (7): (**a**) 700 time steps; (**b**) 900 time steps; (**c**) 1125 time steps.

## Data Availability

Not applicable.

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
