# Peer review of "Traction-Associated Peridynamic Motion Equation and Its Verification in the Plane Stress and Fracture Problems"

_materials, 2023, doi:10.3390/ma16062252_

Round 1

Reviewer 1 Report

The paper treats a special problem in the recently developed approach in solid mechanics named Peridynamic. It may be interesting for the specialists in the field.

It may be useful for the readers who want to expand the knowledge in the new theories in the mechanics of solids.

Author Response

Thank you for your comments!

Reviewer 2 Report

Paper  ID   materials-2220725

Traction-Associated Peridynamic Motion Equation and its verification in The Plane Stress and Fracture Problems

By Ming Yu, Zeyuan Zhou and Zaixing Huang

Submitted to the J Materials

Comments 

The paper deals with the problem of traction forces on boundary surface in peridynamics.

An extended formulation, the traction-associated peridynamic motion equation, is proposed and its compatibility with the conservation laws of linear momentum and angular momentum is demonstrated. Some numerical simulations are proposed and compared with the classical elasticity solutions with which a good agreement has been demonstrated.

The paper presents an interesting topic and fits well with the aim and scope of the journal.

However, before being acceptable for publication in the J. Materials, I would suggest considering some important aspects as detailed below.

- A list of the main symbols used in the manuscript should be introduced.

- The induced body force method adopted in the peridynamics non-local continuum mechanics formulation is based on the so-called transfer function G; it would be important to better describe this function and its physical meaning since it represents the key point of the paper (eq 2). Also, the implications of eq 4 should be described.

- Why are three apexes are used to indicate a point lying on the boundary? This notation is a little weird.

- Eq 6 is written in the deformed configuration of the body (the term on the right-hand side), while on the left-hand side the quantity is referred to the initial configuration. Why? Should also eq 5 be written in the same conditions? If this is the case, please distinguish the volume and boundary of the domain in the reference and current configuration. The same applies to eqs 8, 9.

- The force state vector T shown in eq 7 must be better explained. Sect. 3.2 presents some models for T but they are not sufficiently explained from the mechanical viewpoint.

- A discussion related to the time step amplitude should be included; convergence issue with respect to this aspect is also important to be considered in the manuscript.

- I suggest putting close each other figs 17 and 19; they can be conveniently merged in a single figure for sake of simplicity in comparing the obtained displacement field.

- It is not clearly explained as the damage has been evaluated in sect. 5.4.

- The traction on the boundary surface is dispersed within the domain in the boundary layer with thickness \delta. This feature depends on the function \alpha defined in sect. 3.3. It would be useful to explicitly show this function by varying the involved parameters in a dedicated figure.

- The effect of the size of delta wrt the grid spacing is not considered in the numerical cases presented, it has been assumed to be fixed; I think is it important to study the relative size delta/L and delta/Delta on the displacement and stress results. This aspect is crucial for the treated topic, and a parametric study considering the effect of the boundary forces dispersion distance is fundamental to the proposed extended PD approach.

- Proofread carefully the whole manuscript to fix language errors (e.g. on page 2: “can find in [4,5,8].” should be “can be found in [4,5,8].”, page 4: “internal energy of per unit mass.” Should be “internal energy per unit mass.”, page 6 “Equation (7) is can written as” should be “Equation (7) can be written as”, etc.)

In conclusion, the paper is interesting but it is not sufficiently clearly written and neglects some relevant aspects. I suggest the manuscript be amended according to a minor revision before being accepted for publication in J Materials.

Reviewer 3 Report

This paper proposes a new formulation, called the traction-associated peridynamic motion equation, to address the open question of how to prescribe traction on boundary surfaces in peridynamics. The new formulation introduces an induced body force defined by boundary traction, is compatible with conservation laws, and eliminates the need for volume and surface corrections. Numerical solutions of benchmark examples show that the new formulation is a promising solution for dealing with complex traction boundary conditions.

The paper is within the aim and scope of the journal and appears to be interesting. Before recommending publication, I would like to know the authors' reply to the following points:

1.       In the introduction, "peridynamics" is a noun and should not be preceded by an article. Therefore, please use "peridynamics" instead of "the peridynamics".

2.       I had difficulty understanding how the authors arrived at Eq (4) and would require clarification.

3.       t is still not clear to me, as a reader, how the authors selected the weight function ($\alpha$) from Eq. (22) to Eq. (25), which is the most important function in the proposed work. Furthermore, it is not clear how to select or tune the parameter $k$ upon which the function depends. Can the authors provide additional clarification on these points?

4.       Is it possible to apply the proposed approach to a fully implicit solver rather than the ADR explicit approach? Additionally, could the authors please report the value of damping for each example?

5.       In the section on numerical implementation, it may be valuable for the authors to include a recent reference by Silling and co-authors on their newly developed meshfree scheme in [Computer Methods in Applied Mechanics and Engineering Volume 391, 1 March 2022, 114544].

6.       The authors may want to consider including a reference to [Computational Mechanics (2020) 66:773–793] in the introduction, as it addresses recent studies on the appropriate imposition of boundary conditions in peridynamics without the surface effect problem.

Round 2

Reviewer 3 Report

My comments are addressed. However, at the end of  “Generally, the surface effect needs to be corrected to obtain the correct physical results.”

a reference to the recent publication in [Computer Methods in Applied Mechanics and Engineering Volume 407, 15 March 2023, 115948] be added. After the minor revision, I would recommend the publication.